# Evaluation of Prognostic Factors of Severity in Acute Biliary Pancreatitis

**DOI:** 10.3390/ijms21124300

**Published:** 2020-06-16

**Authors:** Pedro Silva-Vaz, Ana Margarida Abrantes, Sara Morgado-Nunes, Miguel Castelo-Branco, António Gouveia, Maria Filomena Botelho, José Guilherme Tralhão

**Affiliations:** 1Health Sciences Research Centre, University of Beira Interior (CICS-UBI), 6200-506 Covilhã, Portugal; mcbranco@fcsaude.ubi.pt; 2General Surgery Department, Hospital Amato Lusitano, Unidade Local de Saúde de Castelo Branco, 6000-085 Castelo Branco, Portugal; agouveia@ulscb.min-saude.pt; 3Faculty of Health Sciences, University of Beira Interior, 6200-506 Covilhã, Portugal; 4Clinical Academic Centre of Beiras, CACB, 6200-506 Covilhã, Portugal; sara@ipcb.pt; 5Biophysics Institute, Faculty of Medicine, University of Coimbra, 3000-548 Coimbra, Portugal; mabrantes@fmed.uc.pt (A.M.A.); mfbotelho@fmed.uc.pt (M.F.B.); jglrt@hotmail.com (J.G.T.); 6Coimbra Institute for Clinical and Biomedical Research (iCBR) Area of Environment Genetics and Oncobiology (CIMAGO), Faculty of Medicine, University of Coimbra, 3000-548 Coimbra, Portugal; 7CNC.IBILI Consortium/Center for Innovative Biomedicine and Biotechnology (CIBB), University of Coimbra, 3000-548 Coimbra, Portugal; 8Clinical Academic Center of Coimbra, CACC, 3004-561 Coimbra, Portugal; 9Polytechnic Institute of Castelo Branco, Escola Superior de Gestão, 6000-084 Castelo Branco, Portugal; 10Surgery Department, Centro Hospitalar e Universitário de Coimbra (CHUC), University Hospital, Faculty of Medicine, 3000-075 Coimbra, Portugal

**Keywords:** acute biliary pancreatitis, inflammation, prognostic, severity, hepcidin, systemic inflammatory response index

## Abstract

Acute pancreatitis (AP) is an inflammatory disorder of the pancreas that, when classified as severe, is associated with high morbidity and mortality. Promptly identifying the severity of AP is of extreme importance for improving clinical outcomes. The aim of this study was to compare the prognostic value of serological biomarkers, ratios, and multifactorial scores in patients with acute biliary pancreatitis and to identify the best predictors. In this observational and prospective study, the biomarkers, ratios and multifactorial scores were evaluated on admission and at 48 h of the symptom onset. On admission, regarding the AP severity, the white blood count (WBC) and neutrophil–lymphocyte ratio (NLR), and regarding the mortality, the WBC and the modified Marshall score (MMS) showed the best predictive values. At 48 h, regarding the AP severity, the hepcidin, NLR, systemic inflammatory response index (SIRI) and MMS and regarding the mortality, the NLR, hepcidin and the bedside index for severity in AP (BISAP) score, showed the best predictive values. The present study enabled the identification, for the first time, of SIRI as a new prognostic tool for AP severity, and validated hepcidin and the NLR as better prognostic markers than C-reactive protein (CRP) at 48 h of symptom onset.

## 1. Introduction

Acute pancreatitis (AP) is an inflammatory disorder of the pancreas with a multifactorial pathogenesis, in which enzyme activation plays a central role in local pancreatic damage, causing systemic and peripancreatic tissue involvement [1]. In Europe, the incidence ranges from 4.6 to 100 per 100 000 population [2]. Gallstones and alcohol misuse are the main etiologies of AP [2,3]. In the Mediterranean and other southern countries, including Portugal, gallstones is the dominant etiology, with ratios for gallstones:alcohol often more than 5:1 and sometimes more than 10:1 [2]. The overall mortality rate is between 2% and 5% [4,5], but in its severe form it may reach 30% to 50% [6,7,8]. Several studies have shown that the first 48 h after the symptom onset is very important to identify those patients at risk of developing complications or death. This period is also important to outline an appropriate approach based on fluid resuscitation, pain control and nutritional support. Therefore, it is important to implement, as soon as possible, close monitoring or admission in an intensive care unit and to recognize patients who may require referral to other centers [9]. Nevertheless, none of the current biomarkers are consistently accurate, play a definitive role or have widespread applicable value. Actually, early identification is based on multifactorial scores as Ranson, acute physiology and chronic health evaluation (APACHE-II), bedside index for severity in AP (BISAP), systemic inflammatory response syndrome (SIRS) and modified Marshall score (MMS) and several biochemical markers such C-reactive protein (CRP), white blood count (WBC), neutrophils and procalcitonin (PCT)—however all are associated with several limitations [7,10]. Therefore, the early identification of patients at risk of developing severe AP remains a great challenge.

Hepcidin is a circulating peptide hormone that regulates iron concentration in the plasma. The hepcidin levels increase during inflammation, primarily as a result of increased interleukin-6 (IL-6) [11,12]. When AP is present, IL-6 is one of the most important markers in predicting the severity of AP [13]. Previous studies have shown, that hepcidin is synthesized in the liver, kidney, heart, brain and muscles of the skeleton and also by the pancreas [14,15]. Arabul et al. [16] first correlated hepcidin serum levels with the prognosis of AP.

Systemic inflammation markers can be direct or combined, being based on routine, inexpensive and available laboratory tests. Those most described are the neutrophil–lymphocyte ratio (NLR), and the platelet–lymphocyte ratio (PLR). Related to the NLR, the elevation of this ratio during the first 48 h of admission is significantly associated with severe AP, being superior to the total WBC [17]. The PLR has been described as a tool that allows the evaluation of the systemic inflammatory response [18].

The systemic inflammatory response index (SIRI) is a combined marker of systemic inflammation, exclusively used to assess prognosis and therapeutic response in various cancers [19]. SIRI, described for the study of cancer-related inflammation [19], is based on peripheral neutrophil, monocyte and lymphocyte [20]. This is the first study, to our knowledge, that assesses this ratio in a non-malignant inflammatory situation and in acute biliary pancreatitis.

The goal of this study was to evaluate the risk prognostic value of hepcidin, NLR, PLR, SIRI, PCT and BISAP, SIRS and MMS at the early phase of acute biliary pancreatitis.

## 2. Results

### 2.1. Demographic, Clinical and Biochemical Characteristics of Patients

A total of 117 patients with AP were evaluated at recruitment. After applying the exclusion criteria, a total of 75 patients with acute biliary pancreatitis were included in this study. Of them, 47 patients were classified as mild, 15 as moderately severe and 13 as severe AP. The baseline characteristics of these patients are summarized in Table 1. The overall mean age of patients with acute biliary pancreatitis was 72 (24–96) years and the female to male ratio was 1.34:1 (43/32). The comorbidities of the patients with AP included cardiovascular diseases, chronic renal diseases and diabetes. The total mortality rate was 6.7% (5/75). Demographic and clinical characteristics of the patients who died with acute biliary pancreatitis are shown in Appendix A. There was no statistical significance between the different severity groups related to age, gender, body mass index (BMI) and Charlson index (Table 1).

### 2.2. Laboratory Parameters and Score Systems on Admission

Regarding the studied laboratory parameters, the WBC (*p* < 0.001), NLR (*p* < 0.01), and hepdicin (*p* < 0.01) levels of the severe AP group were significantly increased when compared to the mild AP group. No difference was found between the mild and moderately severe AP for WBC, NLR, PCT and hepcidin. The WBC (*p* < 0.05) and NLR (*p* < 0.05) showed significant difference between the moderately severe and severe AP. Regarding the score systems, the statistical significance was verified between the different groups of AP severity for BISAP, SIRS and MMS (*p* < 0.01, *p* < 0.05 and *p* < 0.01, respectively) (Table 1). The BISAP score showed significant difference between the mild and severe AP (*p* < 0.05).

### 2.3. Laboratory Parameters and Score Systems at 48 h after Symptom Onset

When the biochemical markers were evaluated it was found that the WBC, NLR, SIRI, PLR, PCT, CRP and hepcidin levels of the severe AP group were significantly increased compared to the mild AP group (*p* < 0.001 for NLR, SIRI, CRP and hepcidin; *p* < 0.01 for PLR and PCT). Significant difference was found between the mild and moderately severe AP for the WBC, SIRI and CRP (*p* < 0.01) and for the NLR (*p* < 0.05). When the moderately severe and severe AP groups were compared, only hepcidin (*p* < 0.001) and SIRI (*p* < 0.05) showed a significant difference between. When the scores were analyzed, the statistical significance between the different groups of AP severity for BISAP, SIRS and MMS was found (*p* < 0.001) (Table 1). The BISAP, SIRS and MMS showed a significant difference between the mild and severe AP (*p* < 0.001, *p* < 0.01 and *p* < 0.01, respectively) and between the moderately severe and severe AP (*p* < 0.01, *p* < 0.05 and *p* < 0.01, respectively).

### 2.4. Prognostic Accuracy of Laboratory and Score Systems

Regarding the prediction of the severity and mortality of acute biliary pancreatitis, the area under the receiver operating characteristics (ROC AUC) curves for the biochemical parameters and score systems studied are shown in Figure 1 and Figure 2 and in Table 2, Table 3, Table 4 and Table 5.

On admission, the evaluation of the ROC AUC for the severity of acute biliary pancreatitis showed that the WBC, the NLR and BISAP scores had higher areas (Table 2 and Figure 1a). For a WBC level of 14.88 × 10^3^/µL, the sensitivity, specificity, the positive predictive values (PPV) and negative predictive values (NPV) were of 91%, 75%, 44% and 98%, respectively. For a cutoff of 14.64, the NLR showed a sensitivity of 82%, a specificity of 67%, PPV of 36% and NPV of 96% (Table 3). When the mortality was analyzed, the WBC and the MMS showed higher areas (Table 2 and Figure 1b).

When analyzing the ROC AUC for the acute biliary pancreatitis severity, after 48 h of symptom onset, it was found that hepcidin, NLR, SIRI, BISAP and MMS had the highest areas (Table 4 and Figure 2a). Using 154.7 ng/mL as the cutoff point, hepcidin serum concentration at 48 h could readily predict severe acute biliary pancreatitis with a sensitivity of 100%, specificity of 96%, PPV of 86%, NPV of 100% and accuracy of 97%. The sensitivity, NPV and accuracy (97%) for hepcidin were higher than the other commonly used parameters. For a SIRI cutoff of 7.14, the sensitivity, specificity, PPV and NPV was of 82%, 87%, 53% and 96%, respectively (Table 5). When the predictive value for AP mortality was analyzed, the highest ROC AUC were observed for BISAP, NLR and hepcidin (Table 4 and Figure 2b). The hepcidin serum concentration for AP mortality showed the highest accuracy (92%) when compared with the other evaluated parameters.

On admission, it was possible to verify that 17 patients presented values of hepcidin ≥ 72.8 ng/mL, with 29% in patients with mild AP, 24% with moderately severe AP and 47% with severe AP. At 48 h, it was found that 14 patients had hepcidin values ≥ 154.7 ng/mL, with 7% in patients with mild AP, 7% with moderately severe AP and 86% in patients with severe AP.

When hepcidin was compared with other biomarkers and the multifactorial scores studied, it was found that at admission, this biomarker correlated with the PCT at admission (*r* = 0.352, *p* <0.01), PCT at 48 h (*r* = 0.287, *p* < 0.05), SIRI at 48 h (*r* = 0.313, *p* < 0.05) and CRP at 48 h (*r* = 0.493, *p* <0.001), MMS at admission (*p* < 0.05) and MMS at 48 h (*p* <0.05). Regarding hepcidin at 48 h, there was a correlation between the NLR (*r* = 0.654, *p* < 0.001), CRP (*r* = 0.703, *p* < 0.001) and SIRI (*r* = 0.589, *p* < 0.001), all at 48 h. When analyzing multifactorial scores, hepcidin at 48 h correlates with all of the studied scores. When assessing the correlation of hepcidin with revised Atlanta classification (RAC), it was observed that on admission, hepcidin correlates, in mild AP, with WBC at admission (*r* = 0.366, *p* < 0.05) and PCT at admission (*r* = 0.461, *p* <0.01), with no other correlations in both moderately severe and severe AP. At 48 h, hepcidin correlates, in mild AP, with SIRI at 48 h (*r* = 0.346, *p* <0.05) and CRP at 48 h (*r* = 0.536, *p* < 0.001) and in moderately severe AP, with NLR at 48 h (*r* = 0.690, *p* < 0.01), with no correlations with other biomarkers in severe AP.

Regarding SIRI at 48 h, there was a correlation with CRP (*r* = 0.685, *p* < 0.001) and with hepcidin (*r* = 0.589, *p* < 0.001) both at 48 h. When assessing RAC, the correlation was verified with CRP in mild AP (*r* = 0.535, *p* < 0.001) and in moderately severe AP (*r* = 0.611, *p* < 0.01).

## 3. Discussion

AP, an inflammatory disorder of the pancreas, is responsible for a large number of hospital admissions associated with a great economic cost [3,21]. In this sense, the stratification of the severity of AP is crucial, not only for obtaining good clinical outcomes, but also for cost effectiveness. The RAC defines three degrees of severity: mild, moderately severe and severe AP [6]. The severity of AP can be predicted based upon clinical, laboratory, and radiologic risk factors, several severity score systems, and biomarkers. Some of these can be performed on admission to assist in the triage of patients, while others can only be obtained during the first 48 to 72 h or later [7]. The ideal predictor should be rapid, reproducible, inexpensive, minimally invasive and highly accurate [22]. The identification of patients with severe AP, in the first 48 h, through a biomarker, is crucial for an effective approach and good outcomes. Unfortunately, none has to date proven to be a good accurate predictor of the clinical course of AP.

Gallstones are the main cause of AP and result from the impaction of migratory gallstones in the ampulla, causing obstruction and inflammation, being responsible for about 40% of all cases of pancreatitis. Alcohol is the second leading cause of acute pancreatitis, accounting for approximately 20% to 35% of AP cases. Hypertriglyceridemia is the third most common cause of pancreatitis and is responsible for more than 15% of the cases. Post-endoscopic retrograde cholangiopancreatography pancreatitis (PEP) accounts for 1% to 5% of AP cases [23,24].

The population included in this study was exclusively composed of patients with acute biliary pancreatitis, considered in the Mediterranean countries, the most common etiology [2]. Since the gallstones etiology is the most frequent cause of AP in these countries, and although differences are recognized between the various etiologies of AP, there are few studies aimed at this subtype of patient population [25,26].

Etiology can be an important determinant of AP severity. Kamal et al. [23] in their retrospective study found that acute biliary pancreatitis is a more severe disease compared with PEP and alcoholic pancreatitis. Acute biliary pancreatitis is more associated with an extended length of stay, pseudocyst formation, necrosis, organ failure and mortality. One justification may be the association with recurrent episodes of pancreatitis, without definitive treatment, since the repetition of episodes may increase the pancreatic damage with an increased incidence of necrosis. This study showed that SIRS at 48 h is a good predictor of severity and mortality in patients with gallstone etiology. In turn, hypertriglyceridemia AP is associated with local complications and organ failure. Several authors confirmed that this etiology was associated with more AP severity compared to biliary and alcoholic etiology [24].

Although only 10% to 20% of the patients develop severe AP, this degree of severity is associated with a high mortality rate, justified by the progression of the acute inflammatory response to SIRS and/or multiorgan failure [27]. However, the individual response to pancreatic injury is highly variable, and in most cases, unpredictable [28].

According to Párniczky et al. [5], in their prospective and national multicenter study, in a total of 600 adult patient diagnosed with AP, the majority of the patients were mild (61.2%), 30% were moderately severe and 8.8% were severe, when classified with the RAC. These authors suggest that age is a risk factor for the AP severity. In the present study, which included only patients with acute biliary pancreatitis, 62.7% of the patients had mild, 20% had moderately severe and 17.3% had severe AP. The patients had a median age of 72, with the following distribution: mild AP (70 years), moderately severe AP (73 years) and severe AP (74 years), higher than the population studied by Párniczky, which may justify the high rate of severe AP.

For all these reasons, it is urgent to analyze the existing serological markers and multifactorial severity scores and to evaluate and revalidate the markers that can provide an early identification of the patients at risk of developing severe AP and in this subgroup of patients with gallstones etiology. This early assessment of the severity of acute biliary pancreatitis becomes crucial, especially in the first 48 h, since this period is considered important to define the therapeutic approach.

There are currently several biomarkers and multifactorial scoring systems used for the early prediction of severe AP, namely, Ranson, APACHE II, CRP, BISAP and PCT. Miko et al. [29] in their meta-analysis, identified the AUC for several biomarkers and scoring systems to predict the AP severity and mortality. For severity, the AUC for BISAP was 0.79 and for CRP was 0.73; for mortality, the AUC for BISAP was 0.87 and for CRP was 0.73. Párniczky et al. [5], in their prospective and multicenter study, evaluated several biomarkers of prognosis. They observed that the WBC ≥ 23 × 10^3^/µL was associated with severe AP, with significant differences between the three degrees of severity, and the same was verified for CRP (≥200 mg/L) and PCT (≥10 U/L), however, to this last biomarker, the average levels did not differ significantly for the severity of AP.

The originality of the present clinical study is based on the identification and validation of serological markers or calculated ratios for severity and mortality in patients with acute biliary pancreatitis, allowing the identification of which patients are at a higher risk of developing severe AP or the persistence of organ failure and death.

The patients included in this study underwent abdominal ultrasound (US) to confirm the biliary etiology. Patients with etiological doubt after the US underwent endoscopic ultrasonography to exclude microlithiasis and the characterization of the biliary tree.

In this study, several serological biomarkers, ratios and multifactorial scores were evaluated. The ratios were calculated using some of the biomarkers analyzed. The periods evaluated in this study were the admission, considered the ideal time for the assessment of the prognosis and the most used in the majority of studies, and the 48 h of symptom onset, in order to analyze the predictive power of the different markers at the time when persistent organ failure was defined. With this study, it was possible to confirm that serological markers in individual use can be used as early tools for the prognosis of AP severity in the first 48 h after admission. Therefore, on admission, when the severity was assessed, it was found that the serological markers that had the greatest predictive power were the WBC (0.829) and the NLR (0.751). When mortality was assessed, it was found that the WBC (0.886) had the best predictive value. When severity was assessed at 48 h after the onset of symptoms, it was found that hepcidin (0.997), the NLR (0.914) and SIRI (0.906) had the greatest predictive values. In turn, when mortality was analyzed, the biomarkers that showed the greatest predictive power were the NLR (0.963), hepcidin (0.954), CRP (0.929) and SIRI (0.915).

The serum WBC level is elevated as a physiological response to several inflammatory reactions, such as AP. Several studies report it as a good predictive tool for AP severity. However, there are studies that show different and sometimes contradictory results. Serum WBC levels vary with physiological and pathological conditions, including hydration status, stress, and pregnancy [30], that could be responsible for the differences found. Ünal et al. [31] evaluated the role of WBC as a prognostic tool and described for a cutoff of 15.20 × 10^3^/μL, a sensitivity, specificity of 67% and 88%, respectively. Mayer et al. [32] concluded that WBC provided a good discrimination between the mild and severe AP. They also verified that this difference was apparent on the admission day. In turn, Farkas et al. [8] verified that on admission and within 24 h from the onset of symptoms, WBC cannot predict the AP severity and mortality. This is a high-quality study that shows the duality of the results obtained. In the present study, on admission, WBC showed to be a good discriminator between mild and severe AP (*p* < 0.001) and between moderately severe and severe AP (*p* < 0.05). For a cutoff ≥ 14.88 × 10^3^/μL, the sensitivity, specificity and accuracy were 91%, 75% and 79%, respectively. These results are very important since they show a high predictive value for the AP severity at admission. When this marker was evaluated at 48 h of symptom onset, for a cutoff ≥ 12.09 × 10^3^/μL, the sensitivity, specificity and accuracy were of 91%, 80% and 78%, respectively. This study shows that WBC can be an early predictive marker of the severity of acute biliary pancreatitis, easily accessible in clinical practice.

However, several studies have been published focusing on the ratios calculated from hemogram values, which better reflect the systemic inflammatory response, like the NLR, SIRI and the PLR [33].

The NLR allows the measurement of the systemic inflammatory response, since it represents the contrast of two complementary immune pathways, however contradictory. Neutrophils are the main cells responsible for the non-specific inflammatory response and the main initiator of the tissue destructive cascade. In turn, the lymphocyte immune response occurs later and its main function is to mediate and resolve the pre-existing non-specific inflammatory process [34]. Azab et al. [17] assessed the predictive value of the NLR on admission to the ICU and described, for a cutoff value of 4.7, that the sensitivity and specificity were 85.2% and 47.7%, respectively. When applying the cutoff described by these authors in the present study, on admission, a sensitivity and specificity of 100% and 30%, was found, respectively. These authors also demonstrated that the NLR is more stable than serum WBC levels. Gülen et al. [35] demonstrated an increase in the NLR of 9.99 in the patients who died within the first 24 h. In the present study, on admission, for a cutoff value of 14.64, the sensitivity, specificity and accuracy were of 82%, 67% and 73%, while at 48 h, for a cutoff value of 10.89, the sensitivity, specificity and accuracy were of 82%, 85% and 80%, respectively. Kaplan et al. [36] described the cutoff value of 13.64 with 73% sensitivity and 82.7% specificity for the disease severity. This cutoff is similar to that described in the present study on admission. When the predictive value for the mortality of acute biliary pancreatitis was assessed, it was found that, at 48 h after the symptom onset, the NLR had the highest AUC. Like Azab et al., most NLR studies to date have divided the study populations into tertiles. In the present study, the intention was to determine the optimal NLR cutoff value, and this decision may justify the different cutoff value obtained [34]. When analyzing the studies that address the prognostic value of the NLR in AP, there are variations in the cutoff values described [35,36,37,38,39]. Further studies are needed to confirm the role of the NLR as a prognostic tool in pancreatitis, as well as the optimal cutoff value.

SIRI was first described for the study of cancer-related inflammation [40]. This index lists the levels of neutrophils, monocytes and lymphocytes in the blood count of cancer patients and their prognosis, as well as in assessing the response to chemotherapy (QT) regimens [40,41,42]. *SIRI* was defined in the following formula: *SIRI* = *N* × *M*/*L*, where *N* represents neutrophils, *M*—monocytes and *L*—lymphocytes [40]. The optimal cutoff described for the analysis of the response to QT was 2.3 [41]. In the present study, SIRI showed significant changes in the assessment of the acute biliary pancreatitis severity and mortality at 48 h. In assessing the severity, the cutoff ≥ 7.14, showed a sensitivity, specificity, PPV, VPN and accuracy of 82%, 87%, 53%, 96% and 85%, respectively. When the mortality was assessed, it was found that the SIRI had an AUC of 0.915. When this biomarker was correlated at 48 h after the onset of symptoms, with the other parameters, a strong correlation with CRP and a moderate correlation with hepcidin were verified, both at 48 h. In the RAC evaluation and in relation to CRP, SIRI showed a moderate correlation in mild AP, and a strong correlation in moderately severe AP. To our knowledge, this is the first study to assess the applicability of this index in a benign inflammatory process such as AP, hence the cutoff value is different from that described by the malignant pathology studies. Although it is a biomarker with a predictive value only at 48 h, it can be considered an excellent tool using values obtained in the blood count, which are easily applicable in clinical practice.

The PLR has been described as a tool that allows for the measurement of the systemic inflammatory response [43,44]. Platelets play an important role in hemostasis and in the immune-inflammatory response in the context of inflammatory processes [45]. Lymphocytes are involved in the adaptive immune response [45]. In the present study, contrary to what was verified by some authors [45], a change with statistical significance was found for the assessment of acute biliary pancreatitis severity at 48 h. For a cutoff ≥ 187.80, the sensitivity, specificity and accuracy were 73%, 66% and 70%, respectively. Zhou et al. [33] verified, for a cutoff ≥ 169.29, an AUC of 0.621. The authors also confirmed that the PLR was an inflammatory biomarker and the role of platelet as a crucial link between inflammation and microvascular dysfunction. In the present study, the PLR was also considered as a tool with a predictive value for AP severity, although with less accuracy than the NLR and SIRI.

PCT, a 116 amino acid propeptide of calcitonin, was found to appear in high concentration during inflammation and sepsis [46,47]. This biomarker is released by hepatocytes, peripheral monocytes, and G-cells of the thyroid gland. An increased PCT level has been observed in severe AP, pancreatitis necrosis and organ failure [10]. Brunkhorst et al. [48] found a significant difference between the patients with acute biliary pancreatitis and other etiologies. Mofidi et al. [49], in their systematic review, using a cutoff ≥ 0.5 ng/mL, found a sensitivity of 73% and a specificity of 87% for the severity of AP, with an overall AUC of 0.88. In the present study, on admission, for a PCT cutoff ≥ 0.58 ng/mL, the test had a 93% NPV for identifying severe AP patients, with a sensitivity of 73% and a specificity of 65%, and an overall AUC of 0.735. At 48 h of symptom onset, for a cutoff ≥ 2.83 ng/mL, this marker is able to predict severe AP with an NPV, sensitivity and specificity of 91%, 64% and 76%, respectively. These results overlap with those described in the majority of studies confirming the applicability of this marker in predicting the severity of acute biliary pancreatitis.

CRP is an acute-phase protein, described as a predictive tool for AP prognosis and mortality [7]. This acute phase reactant has been widely used as an independent predictor of AP severity and mortality, especially at 48 h after hospital admission [50]. The accepted cutoff value is ≥ 150 mg/dL at 48 h, with a sensitivity, specificity, PPV and an NPV of 80%, 75%, 67% and 86%, respectively [51]. However, as observed in other biomarkers, several studies have shown different results. Farkas et al. [8] found that on admission, CRP shows poor predictive accuracy for AP severity and mortality, but within 24 h from the onset of pain, elevated CRP levels can be used as inclusion criteria in research studies. In the present study, CRP was evaluated as a predictive tool for AP severity at 48 h. Analyzing the cutoff ≥ 150 mg/dL, the sensitivity, specificity, NPV and accuracy found were of 100%, 60%, 100% and 67%, respectively. These study results confirm the predictive power of this marker at 48 h. However, CRP is a marker with a low specificity despite having been widely studied as a prognostic tool in AP [50]. However, it is not AP-specific and its levels should be interpreted according to the patient’s clinical condition [52]. This biomarker presents itself as a useful, easily accessible and low-cost tool with applicability in clinical practice.

Hepcidin was first described by Krause et al. [53], termed as liver-expressed antimicrobial peptide (LEAP-1) with antimicrobial activity. In 2001, Park et al. [54] identified it in urine and renamed it as hepcidin. Hepcidin is a peptide hormone with 25 amino acids, responsible for the regulation of the hemostasis of iron. The hepcidin gene (hepcidin antimicrobial peptide (HAMP)) is located on chromosome 19 [55]. The main source of the synthesis of hepcidin is the liver. Extra-hepatic production of hepcidin was described in the heart, kidney, retina, monocytes and macrophages, alveolar cells, adipocytes, pancreatic β-cells and bile [16,55,56]. The hepcidin levels increase during inflammation, primarily as a result of increased IL-6 and IL-1 [11,57] levels by activating the signal transducer and activator of transcription-3 (STAT3) signaling to hepcidin gene promoter (Figure 3).

Since hepcidin is also synthesized in the pancreas and biliary system, we can state that it predicts the specific and non-specific inflammation of AP, making it a unique biomarker. In the present study, this marker was evaluated on admission, showing a poor predictive power (0.641). However, at 48 h after the symptom onset, it was shown to be the most potent predictive marker for AP severity. For a cut-off ≥ 154.7 ng/mL, the sensitivity, specificity, PPV, NPV and accuracy observed were of 100%, 96%, 86%, 100% and 97%, respectively. Interestingly, when hepcidin on admission was correlated with the biomarkers studied, there was a moderate correlation with CRP at 48 h, while at 48 h there was a strong correlation with CRP and NLR, both at 48 h. When the RAC was assessed, a moderate correlation was verified in mild AP with PCT and PCR at 48 h and a moderate correlation with NLR at 48 h in moderately severe AP. These results can be explained due to some characteristics of hepcidin. Hepcidin is synthesized from the pancreas and released into the systemic circulation via pancreatic beta cells. Another non-exclusive hypothesis could be that the liver, the main source of hepcidin production, better translates the abdominal inflammation related to acute pancreatitis, mediated by IL-1 and IL-6. At 48 h, extra pancreatic hepcidin can be synthesized and released in the bile duct, raising the levels of this marker at a later stage and justifying the greater predictive value in this period of time. Hepcidin, once produced in the pancreas, can be interpreted as a specific marker of pancreatic inflammation. However, hepcidin is a non-specific inflammatory reagent, such as CRP, WBC and PCT.

Both on admission and at 48 h, there was an increase in the serum hepcidin levels in severe AP (47% at admission and 86% at 48 h). This role in the assessment of the prognosis of AP will allow for the early individualized approach to each patient′s clinical status, closer monitoring and possible transfer to the ICU, improving the morbidity and mortality associated with this pathology. The identification of these patients will also allow the evolution of the monitorization of local and systemic complications associated with AP. In the presence of mild AP patients, the laparoscopic cholecystectomy could be performed, during the same hospitalization, in order to eliminate the etiological factor and prevent the recurrence of the pancreatic inflammatory process. The results verified in this study have shown that hepcidin is a better prognostic marker than CRP, being a possible prognostic tool to be adopted in clinical practice.

Regarding the multifactorial prognostic scores, the BISAP score, the SIRS score and the modified Marshall score were evaluated, due to their potential use in a clinical setting and the studies published with their assessment on the prognosis of AP severity.

The BISAP was proposed as a simple-to-use score with clinical data easily available on admission and in the first 24/48 h [59]. Several studies have validated the performance of the BISAP score as a predictive tool for AP severity [60,61]. In its validation, it presented an AUC of 0.83 for mortality (95% CI: 0.8–0.85) [62]. In the present study, when the predictive value for severity was evaluated, the BISAP score presented an AUC of 0.747 at admission and 0.970 at 48 h. For a cutoff ≥ 3, the sensitivity, specificity and accuracy on admission and at 48 h were 45% and 91%, 89% and 89%, 81% and 85%, respectively. When the predictive value for mortality was analyzed, the BISAP score presented an AUC of 0.786 at admission and 0.979 at 48 h. The BISAP score showed the best predictive value, for mortality, at 48 h. Due to its ease of use and calculation, it should be recommended in clinical practice, to evaluate the prognostic of acute biliary pancreatitis, either on admission or at 48 h.

The SIRS score is simple and widely used in a clinical setting, being an important tool for assessing systemic disorders [63]. If SIRS (≥2) persists for more than 48 h after admission, there is an increased risk of developing multiorgan dysfunction, as determined by the MMS [6]. In the present study, both the SIRS score and the MMS had a significant predictive value at 48 h. For a common cutoff of ≥2, they showed a sensitivity, specificity and an accuracy of 82% and 91%, 85% and 93% and 81% and 89%, respectively. They are useful tools for assessing AP severity at 48 h, since it is in this period of time that persistent organ failure is defined.

In summary, this clinical study enabled the identification, for the first time, of SIRI as a new prognostic tool for AP severity, and validated hepcidin and the NLR as better prognostic markers than the CRP at 48 h of symptom onset.

This study, like other studies evaluating the research on AP prognostic biomarkers, has as its main limitation being a single institution study with a small sample size, particularly in moderately severe and severe groups of patients. Further studies are needed to confirm the future applicability of these clinical markers as prognostic factors for acute biliary pancreatitis severity, as they may allow for the early identification of multiorgan failure and acute biliary pancreatitis-associated necrosis and anticipate therapeutic measures that may reduce the risk of associated morbidity and mortality due to this acute pancreatic inflammatory process.

## 4. Materials and Methods

From November 2015 to March 2017, 75 patients with acute biliary pancreatitis were admitted to the Department of General Surgery of the Hospital Amato Lusitano of Unidade Local de Saúde de Castelo Branco, University Teaching Hospital. The protocol was approved by the local ethic committee (reference number 10294/15) and was conducted in compliance with the Declaration of Helsinki. Written informed consent was obtained from each patient.

### 4.1. Study Population

Adults (≥18 years) patients with acute biliary pancreatitis were prospectively recruited. AP was defined according to the revised Atlanta classification (RAC). At least two of the following three features should be present: (1) abdominal pain consistent with AP; (2) serum lipase activity (or amylase activity) at least three times greater than the upper limit of normal value; (3) and characteristic findings of AP on contrast-enhanced computed tomography (CECT) and less commonly on magnetic resonance imaging (MRI) or on transabdominal ultrasonography (US) [6]. Local and systemic complications were defined according to the RAC [6], and systemic inflammatory response (SIRS) was defined according to the established criteria. The severity of AP was defined according to the RAC: mild (no organ or no local or systemic complications), moderately severe (organ failure that was resolved within 48 h and/or the presence of local or systemic complications without persistent organ failure) and severe (presence of persistent organ failure).

### 4.2. Inclusion and Exclusion Criteria

All patients with acute biliary pancreatitis were included. Patients with other causes of AP except biliary, recurrent AP, chronic pancreatitis, pancreatitis due to malignancy, pregnant patients, patients with a time from the onset of disease to presentation in the emergency room greater than 24 h and patients being hospitalized for more than 24 h at the time of recruitment, were excluded from the study. For an etiological confirmation of AP, the patients underwent US with the identification of gallstones or microlithiasis. Patients with etiological doubt after US underwent endoscopic ultrasonography to exclude microlithiasis.

### 4.3. Analysis of Biomarkers

Blood samples were collected from each patient on admission and after 48 h of the symptom onset, drawn into 5 mL heparin-treated tubes and centrifuged for 10 min at 1500× *g*. The serum was then collected and stored at −80 °C until analysis. Seventy-five serum samples were analyzed and categorized into mild AP (*n* = 47), moderately serve (*n* = 15) and severe AP (*n* = 13) on admission and at 48 h after symptom onset. Lipase (U/L), amylase (U/L), CRP (mg/dL), PCT (ng/dL), WBC (10^3^/µL), neutrophil (10^3^/µL), lymphocyte (10^3^/µL), monocyte (10^3^/µL) and platelet were analyzed in accordance with certified standard analysis at the Department of Clinical Pathology, Unidade Local de Saúde de Castelo Branco. The neutrophil/lymphocyte and platelet/lymphocyte ratios were calculated. For the calculation of SIRI, the formula *SIRI* = *N* × *M* ÷ *L* was used. For the evaluation of hepcidin, 9 patients were excluded due to failure in collecting blood samples. Thus, on admission and after 48 h after the symptom onset, 66 serum samples were analyzed: mild AP (*n* = 41), moderately severe (*n* = 13) and severe (*n* = 12) for the evaluation of hepcidin. Serum levels of hepcidin were determined in batch using a commercial Hepcidin 25 (bioactive) HS ELISA kit (DRG Instrument, GmbH, Margurg, Germany) according to the manufacturer’s protocol.

### 4.4. Score Systems Determination

The BISAP score was described in 2008 and consists of a simple method for the early assessment of the mortality risk in patients with AP. It evaluates 5 parameters: BUN, mental status, SIRS, age and pleural effusion. The cutoff used to assess the mortality risk and severity of AP was ≥ 3. In the present study, this score was assessed at admission and at 48 h after symptom onset.

SIRS is a simple score and is widely used in a clinical setting. It evaluates 4 parameters: temperature, respiratory rate, pulse and WBC. This score is defined by the presence of ≥2 of the described parameters. In the present study, this score was assessed at admission and at 48 h after symptom onset.

The MMS allows for the assessment of multiorgan failure in AP. Organ failure includes one or more of the following parameters: cardiovascular, respiratory and kidney failure. Multiorgan failure is present to a cutoff of ≥2. In the present study, this score was assessed at admission and at 48 h after symptom onset.

### 4.5. Statistical Methods

Results are expressed in mean (SD), median (Q1, Q3) or *n* (%). Normality was assessed by means of the Shapiro–Wilk test. Regarding the quantitative variables, the central tendency was compared using the Student t or Mann–Whitney test (2 categories) or ANOVA or Kruskal–Wallis (>2 categories). In the case of qualitative variables, the associations were verified using the chi-square test (χ^2^), with Fisher’s correction when necessary. Receiver operating characteristics (ROC) curves were calculated for assessing the prognostic accuracy and for determining the best cutoff points. A *p* value less than 0.05 indicated statistical significance. Statistical analyses were performed using SPSS 25.0 (SPSS, Inc., Chicago, IL, USA).

## Figures and Tables

**Figure 1 ijms-21-04300-f001:**
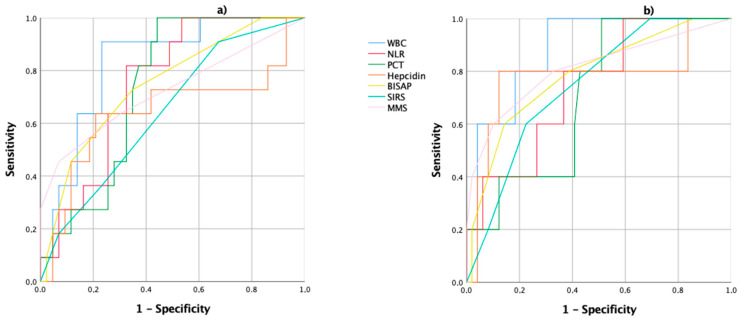
Receiver operating characteristics (ROC) curves of the WBC: white blood count; NLR: neutrophil/lymphocyte ratio; PCT: procalcitonin; hepcidin; BISAP: bedside index of severity in acute pancreatitis; SIRS: systemic inflammatory response syndrome; and the MMS: modified Marshall score, on admission; (**a**) the ROC curves for the prediction of acute biliary pancreatitis severity; (**b**) the ROC curves for the prediction of acute biliary pancreatitis mortality.

**Figure 2 ijms-21-04300-f002:**
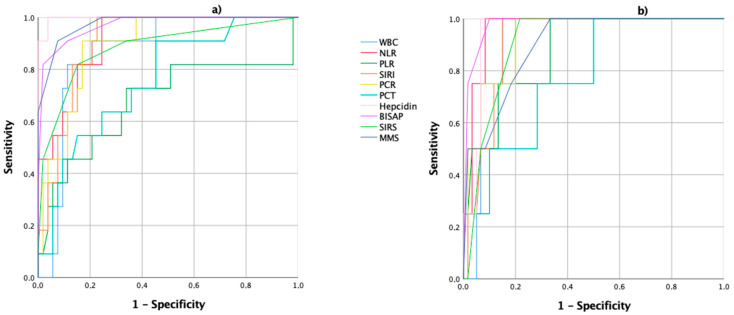
ROC curves of WBC: white blood count; NLR: neutrophil/lymphocyte ratio, SIRI: systemic inflammatory response index; PLR: platelet/lymphocyte ratio; CRP: C-reactive protein; PCT: procalcitonin; hepcidin; BISAP: bedside index of severity in acute pancreatitis; SIRS: systemic inflammatory response syndrome; and the MMS: modified Marshall score, at 48 h of symptom onset; (**a**) the ROC curves for the prediction of acute biliary pancreatitis severity; (**b**) the ROC curves for the prediction of acute biliary pancreatitis mortality.

**Figure 3 ijms-21-04300-f003:**
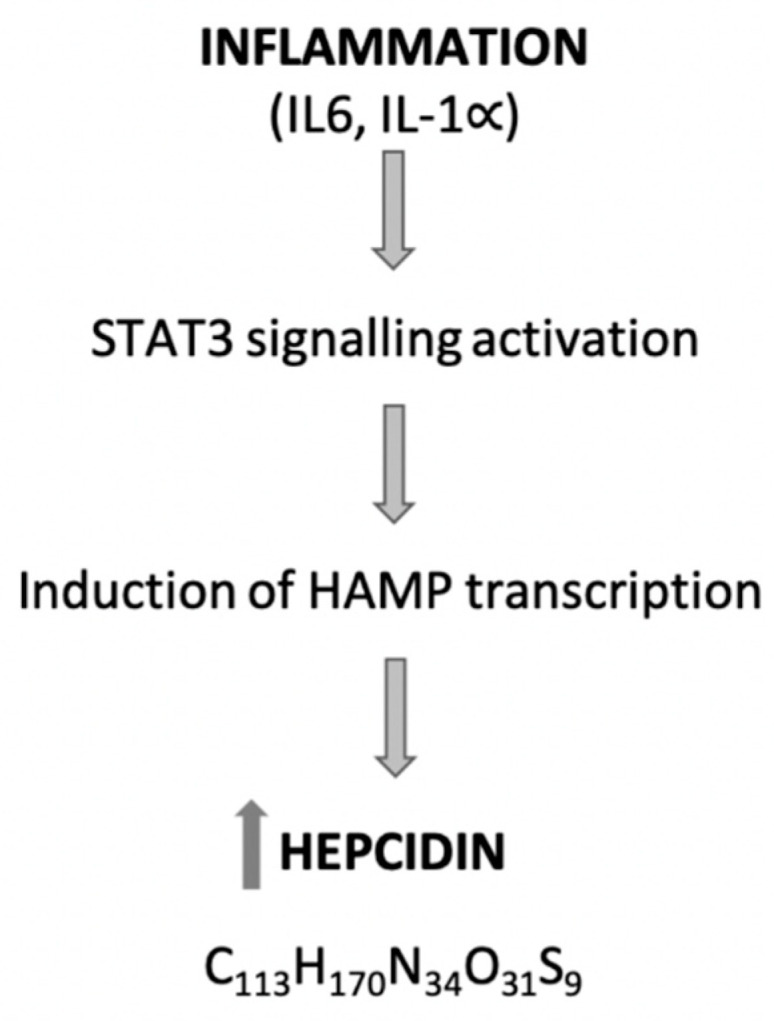
The generation of hepcidin in the inflammatory process. STAT3 activation requires the presence of SAMD4 because its absence prevents STAT3-mediated hepcidin gene expression [58]. This process can be framed within the inflammatory process of AP. IL: interleukin; STAT3: signal transducer and activator of transcription-3; HAMP: hepcidin antimicrobial peptide.

**Table 1 ijms-21-04300-t001:** Baseline characteristics of the patients with acute biliary pancreatitis in the different severity groups.

Variable	Total (*n* = 75)	Mild AP (*n* = 47)	Moderately Severe AP (*n* = 15)	Severe AP (*n* = 13)	*p*
Age (years)	72 ± 17.4	70 ± 18.7	73 ± 14.1	74 ± 17.1	NS
Female sex	57.3%	55.3%	66.7%	53.8%	NS
BMI ≥ 30 kg/m^2^	18.6%	19.1%	20.0%	15.4%	NS
Charlson Index	3 (2–4)	3 (1–4)	3 (2–4)	3 (2–4)	NS
Hospital stay (days)	8 (6–12)	7 (6–9)	11 (7–15)	18 (11–24)	<0.001
Mortality	5 (6.7%)	0	0	5 (38.4%)	NA
**Admission**					
WBC (10^3^/µL)	13.99 ± 4.89	13.03 ± 4.46	12.53 ± 4.76	18.29 ± 4.08	0.001
Neutrophil (10^3^/µL)	11.88 ± 4.74	10.71 ± 4.38	10.79 ± 4.39	16.37 ± 3.55	<0.001
Lymphocyte (10^3^/µL)	1.31 ± 0.91	1.49 ± 1.07	1.04 ± 0.51	1.01 ± 0.52	NS
NLR	13.15 ± 8.50	11.42 ± 8.72	12.36 ± 6.19	18.94 ± 7.86	0.003
PLR	229.11 ± 141.3	215.60 ± 142.5	232.90 ± 122.9	313.73 ± 140.9	NS
SIRI	8.75 ± 8.78	8.84 ± 10.42	6.98 ± 6.30	10.24 ± 5.08	NS
CRP (mg/mL)	59.99 ± 97.66	45.12 ± 77.37	44.50 ± 63.43	118.76 ± 152.31	NS
PCT (ng/mL)	3.41 ± 6.69	2.33 ± 5.58	4.18 ± 5.46	5.77 ± 10.00	0.013
Hepcidin (ng/mL)*	71.21 ± 78.21	52.93 ± 74.43	85.83 ± 57.90	109.76 ± 94.64	0.026
BISAP	1 (1–2)	1 (1–2)	1 (1–3)	2 (1–3)	<0.01
SIRS	1 (0–2)	1 (0–1)	1 (0–2)	1 (1–2)	<0.05
MMS	0 (0–1)	0 (0–0)	0 (0–1)	1 (0–3)	<0.01
**48 h**					
WBC (10^3^/µL)	10.08 ± 4.84	8.60 ± 3.52	11.38 ± 6.69	15.07 ± 2.71	0.001
Neutrophil (10^3^/µL)	8.02 ± 4.89	6.10 ± 3.61	9.37 ± 6.08	13.41 ± 2.63	<0.001
Lymphocyte (10^3^/µL)	1.37 ± 0.58	1.39 ± 0.52	1.25 ± 0.77	0.85 ± 0.36	0.002
NLR	8.26 ± 6.80	5.37 ± 4.27	8.87 ± 4.87	18.06 ± 7.55	<0.001
PLR	174.65 ± 98.19	158.13 ± 83.9	176.43 ± 75.9	232.62 ± 146.7	0.029
SIRI	4.69 ± 4.69	2.94 ± 2.74	5.21 ± 5.39	10.44 ± 5.08	<0.001
CRP (mg/mL)	161.68 ± 134.48	103.9 ± 102.66	200.73 ± 117.49	325.61 ± 106.58	<0.001
PCT (ng/mL)	3.39 ± 5.85	2.01 ± 3.99	4.27 ± 7.03	7.36 ± 8.27	0.001
Hepcidin (ng/mL) *	92.44 ± 102.91	44.96 ± 39.62	74.42 ± 50.78	286.42 ± 81.74	<0.001
BISAP	1 (1–3)	1 (1–2)	2 (1–3)	4 (4–5)	<0.001
SIRS	0 (0–2)	0 (0–1)	0 (0–1)	3 (2–3)	<0.001
MMS	0 (0–1)	0 (0–0)	0 (0–2)	3 (2–4)	<0.001

Results in mean ± SD, median (p25–p75) or *n*(%); * For the evaluation of hepcidin 66 serum samples were analyzed: mild AP (*n* = 41), moderately severe (*n* = 13) and severe (*n* = 12). AP: acute pancreatitis; BISAP: bedside index for severity in acute pancreatitis; BMI: body mass index; CRP: C-reactive protein; MMS: modified Marshall score; NLR: neutrophil–lymphocyte ratio; NA: not applicable; NS: not significant; PCT: procalcitonin; PLR: platelet–lymphocyte ratio; SIRI: systemic inflammatory response index; SIRS: systemic inflammatory response syndrome; WBC: white blood count; *p*: statistical significance for the comparison between the mild, moderately severe and severe categories.

**Table 2 ijms-21-04300-t002:** Area under the receiver operating characteristics curve for the prediction of severe disease and mortality on admission.

Outcome to Predict	Predictor	AUC (95% CI)
Severe disease	WBC	0.829 (0.707–0.951)
	NLR	0.751 (0.615–0.886)
	PCT	0.735 (0.601–0.869)
	Hepcidin	0.641 (0.423–0.859)
	BISAP	0.747 (0.588–0.907)
	SIRS	0.643 (0.469–0.817)
	MMS	0.723 (0.528–0.918)
Mortality	WBC	0.886 (0.767–1)
	NLR	0.743 (0.538–0.948)
	PCT	0.708 (0.512–0.904)
	Hepcidin	0.776 (0.499–1)
	BISAP	0.786 (0.571–1)
	SIRS	0.747 (0.555–0.939)
	MMS	0.810 (0.572–1)

AUC: area under the receiver operating characteristics curve; CI: confidence intervals; MMS: modified Marshall score; NLR: neutrophil/lymphocyte ratio; PCT: procalcitonin; BISAP: bedside index of severity in acute pancreatitis; SIRS: systemic inflammatory response syndrome; WBC: white blood count.

**Table 3 ijms-21-04300-t003:** Prognostic accuracy of the biochemical markers and the score systems of acute biliary pancreatitis severity on admission.

Variable	Se	Sp	PPV	NPV
WBC ≥ 14.88 × 10^3^/µL	91	75	44	98
NLR ≥ 14.64	82	67	36	96
PCT ≥ 0.58 ng/mL	73	65	29	93
Hepcidin ≥ 72.8 ng/mL	64	79	47	90
BISAP ≥ 3	45	89	46	89
SIRS ≥ 2	36	75	24	85
MMS ≥ 2	45	94	56	88

Se: sensitivity; Sp: specificity; PPV/NPV: positive/negative predictive values; MMS: modified Marshall score; NLR: neutrophil/lymphocyte ratio; PCT: procalcitonin; BISAP: bedside index of severity in acute pancreatitis; SIRS: systemic inflammatory response syndrome; WBC: white blood count.

**Table 4 ijms-21-04300-t004:** Area under the receiver operating characteristics curve for the prediction of severe disease and mortality at 48 h.

Outcome to Predict	Predictor	AUC (95% CI)
Severe disease	WBC	0.868 (0.771–0.965)
	NLR	0.914 (0.841–0.988)
	PLR	0.675 (0.464–0.886)
	SIRI	0.906 (0.833–0.979)
	CRP	0.892 (0.807–0.977)
	PCT	0.757 (0.609–0.906)
	Hepcidin	0.997 (0.987–1)
	BISAP	0.970 (0.826–1)
	SIRS	0.883 (0.758–1)
	MMS	0.975 (0.939–1)
Mortality	WBC	0.917 (0.843–0.990)
	NLR	0.963 (0.914–1)
	PLR	0.877 (0.736–1)
	SIRI	0.915 (0.836–0.993)
	CRP	0.929 (0.839–1)
	PCT	0.779 (0.579–1)
	Hepcidin	0.954 (0.901–1)
	BISAP	0.979 (0.943–1)
	SIRS	0.908 (0.824–0.992)
	MMS	0.898 (0.780–1)

AUC: area under the receiver operating characteristics curve; CI: confidence intervals; NLR: neutrophil/lymphocyte ratio; PLR: platelet/lymphocyte ratio; SIRI: systemic inflammatory response index; CRP: C-reactive protein; MMS: modified Marshall score; PCT: procalcitonin; BISAP: bedside index of severity in acute pancreatitis; SIRS: systemic inflammatory response syndrome; WBC: white blood count.

**Table 5 ijms-21-04300-t005:** Prognostic accuracy of the biochemical markers and the score systems of acute biliary pancreatitis severity at 48 h.

Variable	Se	Sp	PPV	NPV
WBC ≥ 12.09 × 10^3^/µL	91	80	43	82
NLR ≥ 10.89	82	85	45	93
SIRI ≥ 7.14	82	87	53	96
PLR ≥ 187.80	73	66	34	93
CRP ≥ 150 mg/mL	100	60	34	100
PCT ≥ 2.83 ng/mL	64	76	38	91
Hepcidin ≥ 154.7 ng/mL	100	96	86	100
BISAP ≥ 3	91	89	55	96
SIRS ≥ 2	82	85	48	94
MMS ≥ 2	91	93	65	97

Se: sensitivity; Sp: specificity; PPV/NPV: positive/negative predictive values; NLR: neutrophil/lymphocyte ratio; SIRI: systemic inflammatory response index; PLR: platelet/lymphocyte ratio; CRP: C-reactive protein; MMS: modified Marshall score; PCT: procalcitonin; BISAP: bedside index of severity in acute pancreatitis; SIRS: systemic inflammatory response syndrome; WBC: white blood count.

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
