# Peer review of "Evaluation of Prognostic Factors of Severity in Acute Biliary Pancreatitis"

_ijms, 2020, doi:10.3390/ijms21124300_

Round 1

Reviewer 1 Report

  • In the introduction you mention that the overall mortality is about 10% in AP. However this is not the case. generally the mortality rate is between 2-5%. Please correct it and cite the relevant articles: PMID: 32328495, PMID: 27798670
  • You mention CRP and WBC as good markers for severity. However the AUC of these markers have strong limitations. Please discuss it accordingly PMID: 31551798
  • The discussion need to be extended as well. Several studies have been published which made severity assessment. For example the one I have mentioned in the introduction. For example a predictive accuracy meta-analysis has been recently showed the differences between the severity scores (PMID: 31507427).
  • Concerning the methods. You most probably is a tercier center in Portugal. The severity rate seems to be extremely high. If yes, please mention it in the methods. If not please to mention other reasons why the AP is so severe in your centre.
  • The most severe form of AP is the HTG induced AP. Please also discuss the severity differences between the different APs (PMID: 32402696)

Author Response

Dear Reviewer and Editor:

Paper number: ijms-820226

Paper title: Evaluation of prognostic factors of severity in acute biliary pancreatitis

Authors: Pedro Silva-Vaz, Ana Margarida Abrantes, Sara Morgado-Nunes, Miguel Castelo Branco, António Gouveia, Maria Filomena Botelho and José Guilherme Tralhão

The authors would like to acknowledge the time spent by the reviewer as well as their valuable comments. We carefully evaluated all the comments and have made changes to improve our manuscript. The changes are in yellow through the text. All the alterations were performed according with reviewer suggestions:

“In the Introduction you mention that the overall mortality is about 10% in AP. However, this not the case. Generally, the mortality rate is between 2-5%. Please correct and cite the relevant articles (PMID: 32328495; PMID: 27798670)”.

Response 1:

Thank you for your comment.

The text has been changed to include the reviewer’s indication as well as the references: Line 46: “The overall mortality rate is between 2% to 5% [4,5]...”

“You mention CPR and WBC as good markers for severity. However, the AUC of these markers have strong limitations. Please discuss it accordingly (PMID: 31551798)”.

Response 2:

Thank you for your comment.

Several articles dedicated to this subject are available, however they evaluate different and heterogeneous populations, with different numbers of sample size and with varied and even contradictory conclusions. Unfortunately, the same is true for CRP and WBC.

Regarding WBC, the studies included in this manuscript (Ünal et al. and Mayer et al.) highlight the advantages of using this biomarker not only the cutoff, but its sensitivity and specificity, being a good discriminator between mild and severe AP. However, it is extremely important to mention that exist studies that show its limitation, namely the work performed by Farkas et al. proposed by the reviewer. These authors developed a large multicentric and multinational, high-quality study in which 1435 patients were enrolled. They found that, on admission, as within 24 hours from the onset of pain, WBC presented poor predictive accuracy, both for severity and mortality. The authors concluded that WBC should not be used either for inclusion criteria or to measure clinical results in research studies.

Regarding CRP, although it is the most used biomarker in the clinical setting, the literature also shows contradictory results. Stirling et al., stipulated the cutoff of 150mg/dL at 48 hours with good sensitivity and specificity. Farkas et al., in its turn, found, on admission, that CRP shows poor predictive accuracy for AP severity and mortality, but within 24 hours from the onset of pain, elevated CRP levels can be used as inclusion criteria in research studies. However, this study did not evaluate CRP at 48 hours, the time described as the best for this biomarker. The present study evaluated CRP at 48 hours after symptom onset, showing a good predictive accuracy for both AP severity (0.892) and mortality (0.929). However, there were no significant changes on admission when comparing the different degrees of severity (p=0.571) nor for mortality (p=0.238).

These contradictory observations may be associated with the different populations under study, with different sample size and different AP etiologies.

The authors are grateful for the availability of the referred article, since it promoted the improvement of the discussion in the sense of the importance of the existence of prognostic biomarkers which may contribute to better outcomes. AP remains a challenging pathology due to its complex pathogenesis which hinders its clinical approach.

The following sentences in yellow were included in the Discussion section:

Line 296: “However, there are studies that show different and sometimes contradictory results. Serum WBC levels vary with physiological and pathological conditions, including hydration status, stress, and pregnancy [30], that could be responsible for the differences found.”

Line 301:“In turn, Farkas et al. [8] verified that on admission and within 24 hours from the onset of symptoms WBC cannot predict the AP severity and mortality. This is a high-quality study that shows the duality of the results obtained.”

Line 379: “However, as observed in other biomarkers, several studies have shown different results. Farkas et al. [8]found that on admission, CRP shows poor predictive accuracy for AP severity and mortality, but within 24 hours from the onset of pain, elevated CRP levels can be used as inclusion criteria in research studies.”

“The discussion needs to be extended as well. Several studies have been published which made severity assessment. For example, the one I have mentioned in the Introduction. For example, a predictive accuracy meta-analysis has been recently showed the differences between the severity scores (PMID: 31507427)”.

Response 3:

Thank you for your comment.

The authors greatly appreciate this valuable comment as well as the suggested bibliographical references. The text has been changed to include the reviewer’s suggestion as well as the references:

Line 265: “There are currently several biomarkers and multifactorial scoring systems used for the early prediction of severe AP, namely, Ranson, APACHE II, CRP, BISAP and PCT. Miko et al. [29] in their meta-analysis identify AUC for several biomarkers and scoring systems to predict the AP severity and mortality. For severity the AUC for BISAP was 0.79 and, for CRP was 0.73, for mortality the AUC for BISAP was 0.87 and for CRP was 0.73. Párniczky et al. [5] in their prospective and multicenter study evaluated several biomarkers of prognosis. They observed that WBC  23x103/µL was associated with severe AP, with significant differences between the three degrees of severity, the same was verified to CRP ( 200mg/L) and PCT ( 10 U/L), however, to this last biomarker, the average levels did not differ significantly between the severity of AP.”

“Concerning the Methods. You most probably are a tercier center in Portugal. The severity rate seems to be extremely high. If yes, please mention it in the methods. If not, please to mention other reasons why the AP is so severe in your centre.”

Response 4:

Thank you for your comment.

The Hospital Amato Lusitano of the Unidade Local de Saúde de Castelo Branco is a level II hospital and is University Teaching Hospital, for this reason more patients seek medical assistance at the Institution. In this study and according to RAC the distribution of acute biliary pancreatitis patients was mild of 67.2%, moderately severe of 20% and severe of 17.3%. Another explanation for the high percentage of critically ill patients is the median age of the population studied (72 years). If the median age of the population is analyzed within the degrees of AP severity the mild is 69 years of age, moderately severe 71 years of age and severe 77 years. Párniczky et al. suggest that age is a risk factor for the severity of AP, and the age group of the population studied may justify the high rate of severe AP observed.

The following sentence in yellow was included in the Discussion section:

Line 252: “According to Párniczky et al. [5] in their prospective and national multicenter study, found that in a total of 600 adult patient diagnosis with AP, the majority of the patients were mild (61.2%), 30% were moderately severe and 8.8% were severe, when classified with the RAC. These authors suggest that age is a risk factor for the AP severity. In the present study, which included only patients with acute biliary pancreatitis, 62.7% of the patients had mild, 20% moderately severe and 17.3% severe AP. The patients had a median age of 72, with the following distribution: mild AP (69 years), moderately severe AP (71 years) and severe AP (77 years), higher than the population studied by Párniczky, which may justify the high rate of severe AP.”

The following sentence in yellow was included in the Methods section:

Line 465: “From November 2015 to March 2017, 75 patients with acute biliary pancreatitis admitted to the Department of General Surgery of the Hospital Amato Lusitano of Unidade Local de Saúde de Castelo Branco, University Teaching Hospital.”

“The most severe form of AP is the HTG induced AP. Please also discuss the severity differences between the different APs (PMID: 32402696)”.

Response 5:

Thank you for your comment.

Several authors report that the AP etiology can be an important determinant of severity of AP. This subject is very important for the substantial improvement of the manuscript since this paper evaluated the AP prognosis in an exclusive population of patients with acute biliary pancreatitis.

The following sentences in yellow were included in the Discussion section:

Line 227: “Gallstones are the main cause of AP and result from the impaction of migratory gallstones in the ampoule, causing obstruction and inflammation, being responsible for about 40% of all cases of pancreatitis. Alcohol is the second leading cause of acute pancreatitis, accounting for approximately 20% to 35% of AP cases. Hypertriglyceridemia is the third most common cause of pancreatitis and is responsible for more than 15% of cases of pancreatitis. Post-endoscopic retrograde cholangiopancreatography pancreatitis (PEP) accounts for 1% to 5% of AP cases [23,24].”

Line 238: “Etiology can be an important determinant of AP severity. Kamal et al. [23] in their retrospective study found that acute biliary pancreatitis is a more severe disease compared with PEP and alcoholic pancreatitis. Acute biliary pancreatitis is more associated with extended length of stay, pseudocyst formation, necrosis, organ failure and mortality. One justification may be the association with recurrent episodes of pancreatitis, without definitive treatment, since the repetition of episodes may increase pancreatic damage with an increased incidence of necrosis. This study showed that SIRS at 48 hours is a good predictor of severity and mortality in patients with gallstone etiology. In turn, hypertriglyceridemia AP is associated with local complications and organ failure. Several authors confirmed that this etiology is associated with more AP severity compared to biliary and alcoholic etiology [24].”

We hope that we have answer all the comments.

Thank you very much for your review.

Best regards

Pedro Silva-Vaz

Department of General Surgery – Unidade Local de Saúde de Castelo Branco

Invited Assistant of Surgery – Faculty of Health Sciences, University of Beira Interior

Unidade Local de Saúde de Castelo Branco

Avenida Pedro Álvares Cabral, s/n

6000-085 Castelo Branco

Portugal

Tel: 00351966498337

Email: pedro.silvavaz@gmail.com

Reviewer 2 Report

In this manuscript, authors aimed to evaluate the prognostic risk value of hepcidin, NLR, PLR, SIRI, PCT and BISAP, SIRS, and MMS at acute biliary pancreatitis, as result hepcidin and SIRI could contribute to its prognosis. This manuscript is unique because the authors found a new role in hepcidin and SIRI. While an interesting study, there are several weaknesses within the study that require clarification and should be addressed prior to publication. I have the following comments.

# Major comments

1.This study was relatively small number of patients evaluating prognosis and mortality of acute biliary pancreatitis. The size of the cohort is too small to substantiate the original observations and consequently generalize the results. It needs to confirm the authors' findings in a larger multicenter study, or in acute pancreatitis, not limited to acute biliary pancreatitis.

  1. Their findings suggested the useful role of hepcidin in the severity of pancreatitis at 48 hours. However,hepcidin levels were also needed to be evaluated at admission to analyze prognosis at the early phase of acute biliary pancreatitis.

  1. It is better to analyze the characteristics of patients with high hepcidin levels, including patients of mild to moderate severity. It is unclear how this distinction would translate into patient outcome, treatment and survival. Furthermore, It would be better to show the correlation between hepcidin and other score classifications and biomarkers.

  1. As a supplemental data, Authors should present detailed data information for the five patients who died of severe pancreatitis.

#Minor comment

  1. Each line of figure 1and 2 are difficult to identify each of them, so you should make them by color.

Author Response

Dear Reviewer and Editor:

Paper number: ijms-820226

Paper title: Evaluation of prognostic factors of severity in acute biliary pancreatitis

Authors: Pedro Silva-Vaz, Ana Margarida Abrantes, Sara Morgado-Nunes, Miguel Castelo Branco, António Gouveia, Maria Filomena Botelho and José Guilherme Tralhão

The authors would like to acknowledge the time spent by the reviewer as well as their valuable comments. We carefully evaluated all the comments and have made changes to improve our manuscript. The changes are in blue through the text. All the alterations were performed according to reviewer suggestions:

“This study was relatively small number of patients evaluating prognosis and mortality of acute biliary pancreatitis. The size of the cohort is too small to substantiate the original observations and consequently generalize the results. It needs to confirm that authors’ findings in a larger multicenter study, or in acute pancreatitis, not limited to acute biliary pancreatitis”.

Response 1:

Thank you for your comment.

As mentioned in the discussion section, the authors assume as the main limitation of this study the sample size, relating to the fact that it is a uni-institutional study and with patients exclusively with biliary etiology.

The authors intend to initiate a multicenter and national study, pioneering in Portugal, to assess the prognosis, not only in patients with lithiasis etiology, but encompassing all etiologies, in order to confirm the observations found and, consequently, generalize the results.

However, the authors consider that the results observed in this study are important for improving the acute biliary pancreatitis approach and the outcomes, with biomarkers that can be easily applied in the clinical setting.

The following sentence in yellow was included in the Discussion section:

Line 456: “This study, like other studies evaluating on research of AP prognostic biomarkers, has as its main limitation being a single Institution study with a small sample size, particularly in moderately severe and severe group of patients.”

“Their findings suggested the useful role of hepcidin in the severity of pancreatitis at 48 hours. However, hepcidin levels were also needed to be evaluated at admission to analyze prognosis at the early phase of acute biliary pancreatitis”.

Response 2:

Thank you for your comment.

This subject was discussed during the study design. The authors decided to evaluate the role of hepcidin prognostic marker at 48 hours, since it allows for a comparison with the biochemical marker currently accepted in clinical settings, the CRP and that, as we know at admission, has a poor predictive value of severity and mortality, but at 48 hours and with a cutoff of 150mg/dL it assumes an extremely important role as a marker of therapeutic approach decision, and also because 48 hours is the defined time to assume if organ failure is defined as transient or persistent.

The reviewer's suggestion is very pertinent and important, and the authors decided to include the results obtained for hepcidin on admission in this manuscript.

In this sense, tables 1 (Line 96), 2 (Line 164) and 4 (Line 180) and figure 1 (Line 151) were changed to include this new information.

The following sentence in yellow was included in the Results (2.2) section:

Line 107: “Regarding the studied laboratory parameters, the WBC (p<0.001), NLR (p<0.01) and hepcidin (p<0.01) levels of the severe AP group were significantly increased when compared to the mild AP group. No difference was found between mild and moderately severe AP for WBC, NLR, PCT and hepcidin.”

The following sentence in yellow was included in the Discussion section:

Line 405: “In the present study, this marker was evaluated on admission showing a poor predictive power (0.641).”

“It is better to analyze the characteristics of patients with high hepcidin levels, including patients to mild to moderate severity. It is unclear how this distinction would translate into patient outcome, treatment and survival. Furthermore, it would be better to show the correlation between hepcidin and other scores classifications and biomarkers”.

Response 3:

Thank you for your comment.

The authors evaluated the distribution of serum hepcidin values and found that the highest percentage of patients with the highest cutoff values studied were in the severe AP. This finding is important since it will allow for the identification of patients at risk of developing severe AP, and help to decide on individualized therapeutic approaches which may result in improved outcomes and decreased the mortality associated to AP.

The following sentences in yellow were included in the Results section:

Line 196: “On admission, its was possible to verify that 17 patients presented values of hepcidin ≥ 72.8ng/mL, 29% in patients with mild AP, 24% in moderately severe AP and 47% in severe AP. At 48 hours, it was found that 14 patients had hepcidin values ≥ 154.7ng/mL, with 7% in patients with mild AP, 7% in moderately severe AP and 86% in patients with severe AP.”

Line 197: “When hepcidin was compared with other biomarkers and multifactorial scores studied, it was found that at admission this biomarker correlated with PCT at admission (r = 0.279, p <0.05), NLR at 48 hours (r = 0.383, p <0.01), PLR at 48 hours (r = 0.352, p <0.01) and PCR at 48 hours (r = 0.461, p <0.001), MMS at admission (p <0.05) and MMS at 48 hours (p <0.05). Regarding hepcidin at 48 hours, there was a correlation between NLR (r = 0.678, p <0.001), PCR (r = 0.646, p <0.001) and SIRI (r = 0.601, p <0.001) all at 48 hours. When analyzing multifactorial scores, hepcidin at 48 hours correlates with all of the studied scores. When assessing the correlation of hepcidin with RAC, it was observed that on admission hepcidin correlates, in mild AP, with WBC at admission (r = 0.366, p <0.05) and PCT at admission (r = 0.461, p <0.01), with no other correlations in both moderately severe and severe AP. At 48 hours, hepcidin correlates, in mild AP, with SIRI at 48 hours (r = 0.346, p <0.05) and CRP at 48 hours (r = 0.536, p <0.001) and in moderately severe AP with NLR at 48 hours (r = 0.690, p <0.01), with no correlations with other biomarkers in severe AP.”

Line 212: “Regarding SIRI at 48 hours, there was a correlation with CRP (r = 0.685, p <0.001) and with hepdicin (r = 0.589, p <0.001) both at 48 hours. When assessing RAC, correlation was verified with CRP in mild AP (r = 0.535, p <0.001) and in moderately severe AP (r = 0.611, p <0.01).”

The following sentences in yellow were included in the Discussion section:

Line 409: “Interestingly, when hepcidin on admission was correlated with the biomarkers studied, there was a moderate correlation with CRP at 48 hours, while at 48 hours there was a strong correlation with CRP, NLR and SIRI all at 48 hours. When RAC was assessed, a moderate correlation was verified in mild AP with PCT and PCR at 48 hours and a moderate correlation with NLR at 48 hours in moderately severe AP.”

Line 422: “Both on admission and at 48 hours, there was an increase in serum hepcidin levels in severe AP (47% at admission and 86% at 48 hours). This role in the assessment of the prognosis of AP will allow for the early individualized approach to each patient's clinical status, closer monitoring and possible transfer to the ICU, improving the morbidity and mortality associated with this pathology.  The identification of these patients will also allow evolution monitorization of local and systemic complications associated with AP. In the presence of mild AP patients, the laparoscopic cholecystectomy could be performed, during the same hospitalization, in order to eliminate the etiological factor and prevent recurrence of the pancreatic inflammatory process.”

“As a supplemental data, authors should present detailed data information for the five patients who died of severe pancreatitis.”

Response 4:

Thank you for your comment.

The authors included as supplementary materials the Table S1 which presents the demographic and clinical characteristics of the patients that died with severe AP. (Lines 88 and 747)

“Each line of figure 1 and 2 are difficult to identify each of them, so you should make them by color”.

Response 5:

Thank you for your comment.

The authors changed Figure 1 (Line 151) and 2 (Line 157), related to the ROC curves and assigned colors to the different markers and scores under study in order to facilitate their reading.

We hope that we have answer all the comments.

Thank you very much for your review.

Best regards

Pedro Silva-Vaz

Department of General Surgery – Unidade Local de Saúde de Castelo Branco

Invited Assistant of Surgery – Faculty of Health Sciences, University of Beira Interior

Unidade Local de Saúde de Castelo Branco

Avenida Pedro Álvares Cabral, s/n

6000-085 Castelo Branco

Portugal

Tel: 00351966498337

Email: pedro.silvavaz@gmail.com

Reviewer 3 Report

In this wise and well written article authors analyze the predictive values of several scoring systems and biomarkers for acute pancreatitis, an important issue in clinical medicine.

They demonstrate the high accuracy of SIRI, hepcidin and neutrophil to lymphocyte ratio for prediction of severe cases of acute pancreatitis.

Minor comments to improve the manuscript 

Line 139 : highest instead of higher

Line 190 : radiologic instead of imgiologic

Line 339 : Beside the role of pancreas synthesis hypothesized by authors in the increase of circulating hepcidin, another non-exclusive hypothesis could be that liver (the main site of hepcidin production in humans) better translate the abdominal inflammation related to acute pancreatis (mediated by IL1 and IL6). 

Line 397 : the second sentence has to be rewritten : I guess that these patients were excluded from the study.

Author Response

Dear Reviewer and Editor:

Paper number: ijms-820226

Paper title: Evaluation of prognostic factors of severity in acute biliary pancreatitis

Authors: Pedro Silva-Vaz, Ana Margarida Abrantes, Sara Morgado-Nunes, Miguel Castelo Branco, António Gouveia, Maria Filomena Botelho and José Guilherme Tralhão

The authors would like to acknowledge the time spent by the reviewer as well as their valuable comments. We carefully evaluated all the comments and have made changes to improve our manuscript. The changes are in yellow through the text. All the alterations were performed according with reviewer suggestions:

“Line 139: highest instead of higher”.

Response 1:

Thank you for your comment.

The text has been changed to include the reviewer’s suggestion.

The following sentence in yellow was included in the Results section:

Line 147: “Hepcidin serum concentration for AP mortality showed the highest accuracy (92%) when compared with the other evaluated parameters”

“Line 190: radiologic instead of imagiologic”.

Response 2:

Thank you for your comment.

The text has been changed to include the reviewer’s suggestion.

The following sentence in yellow was included in the Discussion section:

Line 219: “The severity of AP can be predicted based upon clinical, laboratory, and radiologic risk factors, several severity score systems, and biomarkers.”

“Line 339: Beside the role of pancreas synthesis hypothesized by authors in the increase of circulating hepcidin, another non-exclusive hypothesis could be that liver (the main site of hepcidin production in humans) better translate the abdominal inflammation related to acute pancreatitis (mediated by IL1 and IL6)”.

Response 3:

Thank you for your comment.

The hypothesis mentioned by the reviewer was included in the text. The authors greatly appreciate this valuable observation.

The following sentence in yellow was included in the Discussion section:

Line 415: “Another non-exclusive hypothesis could be that the liver, the main source of hepcidin production, better translates the abdominal inflammation related to acute pancreatitis, mediated by IL1 and IL6.”

“Line 397: the second sentence has to be rewritten: I guess that these patients were excluded from the study”.

Response 4:

Thank you for your comment.

The text has been changed to include the reviewer’s suggestion.

The following sentence in yellow was included in the Materials and Methods (4.2) section:

Line 488: “Patients with other causes of AP except biliary, recurrent AP, chronic pancreatitis, pancreatitis due to malignancy, pregnant patients, patients with time from onset of disease to presentation in the emergency room greater than 24 hours and patients being hospitalized for more than 24 hours at the time of recruitment, were excluded from the study.”

We hope that we have answer all the comments.

Thank you very much for your review.

Best regards

Pedro Silva Vaz

Department of General Surgery – Unidade Local de Saúde de Castelo Branco

Invited Assistant of Surgery – Faculty of Health Sciences, University of Beira Interior

Unidade Local de Saúde de Castelo Branco

Avenida Pedro Álvares Cabral, s/n

6000-085 Castelo Branco

Portugal

Tel: 00351966498337

Email: pedro.silvavaz@gmail.com

Round 2

Reviewer 2 Report

The authors did a commendable job in addressing reviewer concerns and the manuscript has significantly improved as compared to the original version. No further questions nor comments for the authors is remained.